# Human Neutrophil α-Defensins 1–3 Are Upregulated in the Microenvironment of Fibrotic Liver

**DOI:** 10.3390/medicina59030496

**Published:** 2023-03-02

**Authors:** Rami Abu Fanne, Emad Maraga, Eiass Kassem, Gabriel Groisman, Naama Amsalem, Abdel-Rauf Zeina, Moran Abu Mouch, Randa Taher, Saif Abu-Mouch

**Affiliations:** 1Department of Cardiology, Hillel Yaffe Medical Center Affiliated with Rappaport Faculty of Medicine, Technion—Israel Institute of Technology, Haifa 3200003, Israel; 2Department of Clinical Biochemistry, Hadassah Hebrew University Hospital, Jerusalem 91120, Israel; 3Department of Pediatrics, Hillel Yaffe Medical Center, Hadera 38100, Israel; 4The Institute of Pathology, Hillel Yaffe Medical Center Affiliated with Rappaport Faculty of Medicine, Technion—Israel Institute of Technology, Haifa 3200003, Israel; 5Department of Radiology, Hillel Yaffe Medical Center, Hadera 38100, Israel; 6Department of Internal Medicine B, Hillel Yaffe Medical Center Affiliated with Rappaport Faculty of Medicine, Technion—Israel Institute of Technology, Haifa 3200003, Israel

**Keywords:** fatty liver, inflammation, alpha defensin, fibrosis, neutrophil activation

## Abstract

*Background and Objectives*: Neutrophil infiltration is an established signature of Non-Alcoholic Fatty Liver Disease (NAFLD) and Steatohepatitis (NASH). The most abundant neutrophilic peptide, alpha-defensin, is considered a new evolving risk factor in the inflammatory milieu, intimately involved in lipid mobilization. Our objective is to assess for potential association between alpha-defensin immunostains and NAFLD severity. *Materials and Methods*: We retrospectively investigated the liver biopsies of NAFLD/NASH patients, obtained at Hillel Yaffe Medical center between the years 2012 and 2016. Patients’ characteristics were recorded, including relevant blood tests at the time of biopsy. Each biopsy was semi-quantitatively scored using NAFLD Activity Score (NAS) and NASH fibrosis stage. The biopsies were immunostained for alpha-defensin. The precipitation of alpha-defensin was correlated to NAS and fibrosis. *Results*: A total of 80 biopsies were evaluated: male ratio 53.2%, mean age 44.9 ± 13.2 years, 54 had fibrosis grades 0–2, and 26 were grade 3–4. Conventional metabolic risk factors were more frequent in the high-grade fibrosis group. Immunostaining for alpha-defensin disclosed higher intensity (a.u.) in grade 3–4 fibrosis relative to grades 0–2, 25% vs. 6.5%, *p* < 0.05, respectively. Moreover, alpha-defensin staining was nicely co-localized with fibrosis. *Conclusions*: In our group of NASH/NAFLD patients, higher metabolic risk profile was associated with higher fibrosis grade. Immunostaining for alpha-defensin showed patchy intense staining concordant with high fibrosis, nicely co-localized with histological fibrosis. Whether alpha-defensin is a profibrotic risk factor or merely risk marker for fibrosis must be clarified in future studies.

## 1. Introduction

NAFLD is largely considered a hepatic manifestation of the metabolic syndrome, as its prevalence aligns with the main correlates of the syndrome. The pathogenesis of NAFLD is partially elucidated. NAFLD is believed to represent a continuum of hepatic injuries, ranging from simple steatosis to NASH. The vast majority of NAFLD patients have a benign clinical course, 25% develop NASH, and one third of them end with cirrhosis [1,2,3]. The precise mechanisms underlying the pathogenesis of progression from NAFLD to NASH are partially understood. The pathogenesis of NAFLD was initially dominated by the “two-hit” paradigm; [4], the “first hit” being hepatocytes’ lipid saturation in association with insulin resistance, and the “second hit” representing a constitution of multiple inflammatory insults. However, successive studies challenged this concept, and the complexity of NAFLD was better explained by the “multiple hit model”: multiple, parallel injuries acting together to induce NAFLD including hepatic steatosis, hepatocyte lipotoxicity, immune-mediated inflammation, mitochondrial dysfunction and associated oxidative stress, and disruption of the circadian clock. In this regard, accumulating evidence has suggested a key role of neutrophils in the development and progression of NAFLD [5,6]. On the one hand, neutrophil infiltration is a critical pathologic feature of hepatic injury. Experimental animal model of steatohepatitis uncovered neutrophil–hepatic stellate cell interactions that promoted liver fibrogenesis [7], and the major neutrophilic enzyme, myeloperoxidase, was intimately linked with hepatic steatosis and fibrosis progression [8,9]. On the other hand, neutrophils actively alleviated hepatic inflammation and fibrosis in different animal model [10]. Hence, the net hepatic effect of neutrophils is still elusive. In attempt to better clarify the relationship between neutrophils and NAFLD we focused herein on the most abundant neutrophilic peptide, the alpha defensin. Alpha defensin is a critical component of the first line defense arm of the innate immunity. We have previously described the effect of alpha-defensin on low density lipoprotein (LDL) shuttling, and atherogenesis using transgenic mice expressing human alpha-defensin in their polymorphonuclear leukocytes (Def^+/+^). Def^+/+^ mice developed [alpha-defensin·LDL] dimers that accelerated the clearance of LDL from the circulation, augmented hepatic, and vascular deposition and retention of LDL, induced endothelial cathepsins, and increased the development of lipid streaks in the aortic roots [11]. We have recently demonstrated anti-steatosis and profibrotic net effects of alpha-defensin in the same mouse line (unpublished data). Similar findings were described earlier [12] when pan-tissue expression of alpha-defensin increased hepatic fibrosis by inducing hepatic stellate cell proliferation, with a null effect on hepatic steatosis.

In the current paper, we tested human NASH/NAFLD liver slides for alpha-defensin staining. We found a substantial association between alpha-defensin intensity and fibrosis grade.

## 2. Materials and Methods

### 2.1. Study Population

In a retrospective study, 80 patients with presumptive diagnosis of NAFLD (Liver biopsy proven) followed at the liver clinic of Hillel Yaffe medical center between 2012 and 2016 were included. The patients recruited had no history of excessive alcohol consumption, other causes of liver disease (autoimmune, genetic, or endocrine diseases, hepatocellular carcinoma, HCV, HBV and/or HIV infection), or overt systemic inflammatory disease, and none was using regular anti-inflammatory medications.

Patients’ related formalin-fixed and paraffin-embedded liver samples were retrieved from the histopathology laboratory of Hillel Yaffe medical center. Relevant biochemical parameters drawn just before each liver biopsy were retrieved from the Hillel Yaffe medical center archive. All tests were previously analyzed using the Cobas^®^ 6000 analyzer series (Roche Diagnostics).

This study has been reviewed and approved by the Ethic Board of Hillel Yaffe medical center and was in accordance with the declaration of Helsinki. This is a retrospective study, so an informed consent was not required.

Patients were divided into two groups according to fibrosis score of 0–2 or 3–4.

### 2.2. Human Livers Staining for Bland Steatosis, Fibrosis, and Alpha-Defensin

Fresh histology slides were obtained from formalin fixed paraffin embedded blocks of NASH/NAFLD liver biopsies, and different staining protocols were applied:H&E for morphology;Masson trichrome blue for fibrosis;immunohistochemical staining for alpha-defensin.

The NAFLD Activity Score (NAS) and NASH fibrosis stage (0–4) were applied. The NAS ranges from 0 to 8, the sum of scores of steatosis (0–3), lobular inflammation (0–3), and hepatocyte ballooning (0–2). According to the NASH CRN system, fibrosis stage 0 = no fibrosis; stage 1 = centrilobular pericellular fibrosis; stage 2 = centrilobular and periportal fibrosis; stage 3 = bridging fibrosis; and stage 4 = cirrhosis [13].

For alpha-defensin immunostaining, liver sections (10µm) were incubated with a monoclonal antibody against alpha-defensin (HycultBiotech) and stained by the avidin–biotin complex (ABC) procedure with diamino-benzidine (DAB) as substrate. The primary antibody was replaced by the same concentration of irrelevant immunoglobulin as a negative control. Positive staining was visualized by the brown-colored [3,3-di-aminobenzidine] (DAB) reaction product.

### 2.3. Quantitative Analysis

Immunohistochemical reactions were carried out on three sections. Five randomly selected microscopic fields (each field 0.785 mm^2^, 200 × magnification) from each liver section were then photographed with a digital camera (Olympus C-4000, Tokyo, Japan). The data were subjected to software-assisted image analysis (ImageJ^®^) for quantification of immunohistochemical staining intensity. Comparisons were then made between the three different stains to evaluate association/colocalization between alpha-defensin immunostaining and fibrosis in each specimen.

### 2.4. Statistical Analyses

Baseline categorical variables were summarized using proportion, and continuous variables were summarized using mean ±SD. Student’s *t*-test for unpaired data was used to compare continuous variables. The Fisher’s exact test was used for categorical variables. All analysis was performed using SPSS software version 13 (SPSS Inc., Chicago, IL, USA). Differences were considered statistically significant if *p* < 0.05.

## 3. Results

Liver biopsy samples performed between 2012 and 2016 from 80 patients with NAFLD/NASH were included in this study. The Arab: Jewish ratio was 50:30 (53% men) and their mean age (S.D.) was 44.9 ± 13.2 years and mean BMI was 31.2 ± 3.8 kg/m^2^. Not surprisingly, the prevalence of the classical metabolic risk factors for fatty liver was high among the study group: 37.8% diabetes, 32.4% hypertension, and 47.9% with hyperlipidemia (Table 1). Nevertheless, the documented cardiovascular manifestations, including CAD and CHF, were low (1.37% each), which is probably a reflection of the mean young age of the study group. The biochemical tests revealed normal total bilirubin, alkaline phosphatase, albumin, and coagulation parameters, but AST and ALT were elevated at values 50.2 ± 31.4 U/L and 75± 46.7 IU/L, respectively. The mean baseline levels of the inflammatory risk marker (± S.D.) CRP was 9.7 ± 13 mg/L.

We further tested baseline metabolic and biochemical parameters as a function of the obtained histological fibrosis score: 0–2 or 3–4 scores (Table 2). Advanced fibrosis was associated with significantly higher mean BMI and plasma CRP values, and lower serum ALT and albumin levels. 

### Assessment of Liver Biopsy Fragments for NAS, NASH Fibrosis Stage, and Alpha-Defensin Immunostaining

Liver biopsies were evaluated by expert pathologist. Each biopsy was given a NAS (0–8) and NASH fibrosis score (0–4). Patients were divided into two groups according to NASH fibrosis scores of 0–2 (53 patients) and 3–4 (27 patients). The baseline patients’ characteristics according to fibrosis grouping, including the percentage of hepatic injury/hepatitis, indicated through the ALT blood test are shown in Figure 1. One can appreciate that the prevalence of diabetes, hypertension, obesity, and hyperlipidemia were significantly higher in the high fibrosis score group. Interestingly, the percentage of abnormal liver enzymes/hepatitis was significantly lower in this group. On the other hand, Figure 2 depicts the different components of the NAS score as a function of fibrosis severity. Carefully scrutinizing the data disclose significantly more tissue inflammation in the high fibrosis score group, a trend to higher NASH score (4.5 vs. 3.96, *p* = 0.09), and the degree of steatosis as not associated with the fibrosis stage. Interestingly, a positive association was found between the intensity of alpha defensin immunostaining and the severity of liver fibrosis (Figure 3G). Moreover, heterogeneous staining for released alpha-defensin in the absence of neutrophils was detected in association with areas of tissue scaring (Figure 3A–D).

## 4. Discussion

Non-alcoholic fatty liver disease (NAFLD) is the leading cause of chronic liver disease, with 25% of the adult population affected [14]. While infiltration of neutrophils is a common feature of patients as well as mice with fatty liver diseases [15,16], the role of neutrophils during liver fibrogenesis remains controversial. On the one hand, several studies reported profibrotic effects of neutrophils [17,18,19]. On the other hand, neutrophils were mechanistically associated with alleviation of fibrosis through matrix metalloproteinase-induced collagen degradation [20,21], or polarizing macrophages toward anti-inflammatory phenotype [22].

As the major neutrophilic peptide, alpha defensin is an emerging metabolic risk factor, intimately linked to atherogenesis. Moreover, it exerts an active role in liver fat deposition, including the formation of neutrophil extracellular traps [23].

In this series of NAFLD liver biopsies, we provide evidence for anatomical and semiquantitative association between the major neutrophilic peptide, alpha defensin, and fibrosis severity. Ibusuki et al. [12] demonstrated a profibrotic effect of alpha defensin overexpression; they also displayed a lesser fat content in their model that was ascribed to advanced liver disease/cirrhosis. Of note, we found in a previous study, involving physiological overexpression of alpha defensin, lesser fat content in alpha defensin transgenic mice (unpublished data), despite the lack of markers/features of advanced liver disease. However, in the current study, steatosis was non-significantly related to fibrosis level (Figure 2) and alpha defensin intensity. In line with previous reports, the traditional risk factors for NAFLD and the predisposition of individuals to cirrhosis, hypertension, hyperlipidemia, diabetes, and obesity were more prevalent in patients with more advanced fibrosis [24].

The current described overexpression of alpha-defensin may reflect its cumulative inflammatory contribution to fibrogenesis as a “footprint” of neutrophils. The multiple hit model and the characterization of diverse mechanisms for NAFLD pathogenesis were clinically translated into potential therapeutic agents, which are currently under clinical evaluation [25]. Beyond the classical risk factors/hits for liver injury, alpha defensin may evolve as an innovative risk factor for liver fibrosis, which may potentially serve as potential therapeutic target, or a marker for prognostic assessment. Colchicine is a classical neutrophil stabilizing agent; among other things, it reduces alpha defensin levels [11]. In chronic liver disease patients, long-term colchicine application was proved as safe and effective antifibrotic strategy [26]. Hence, risk stratification of NAFLD patients according to alpha defensin-staining level might be helpful in managing this population.

Overall, the results demonstrate elevation of alpha defensin in fibrotic liver areas, an association that may reflect a cause-and-effect relationship or a marker of prognostic assessment. This observation might be an avenue for future research with a potential impact on clinical practice.

## Figures and Tables

**Figure 1 medicina-59-00496-f001:**
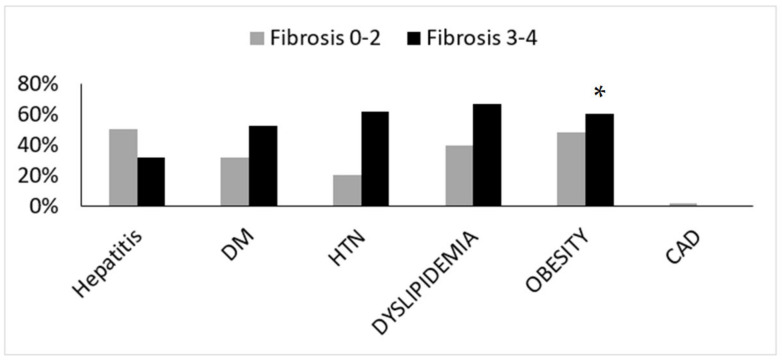
Baseline patients’ characteristics according to the fibrosis score. DM—diabetes mellitus, HTN—hypertension, CAD—coronary artery disease. (*) indicates *p* < 0.05.

**Figure 2 medicina-59-00496-f002:**
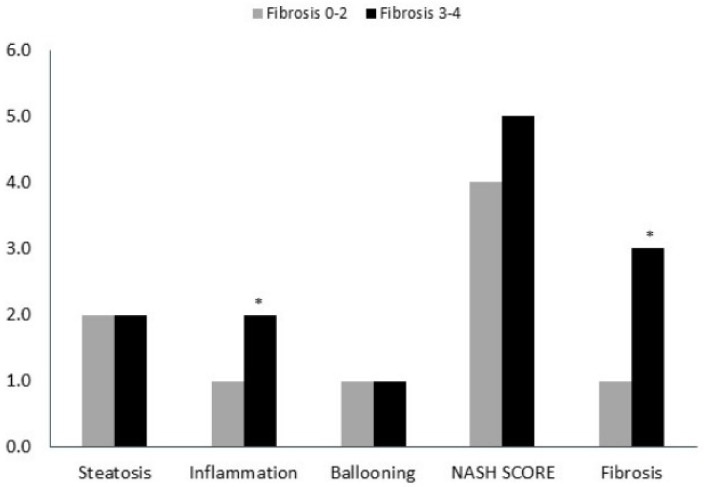
Different median components of the NAFLD Activity Score (0–8) as a function of fibrosis severity. (*) indicates *p* value < 0.05.

**Figure 3 medicina-59-00496-f003:**
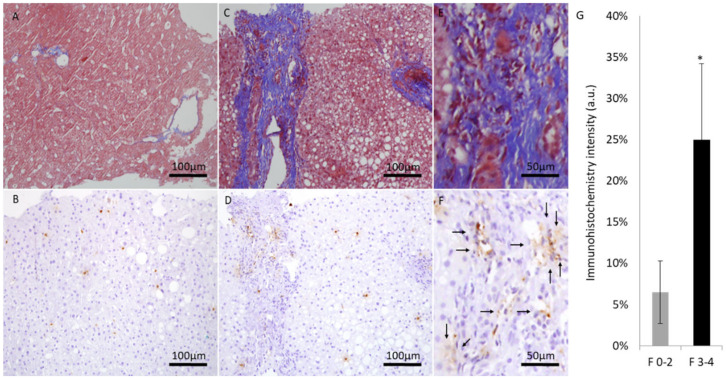
Representative liver sections at ×200 magnification (scale bar, 100 μm) and ×400 magnification (scale bar, 50 μm). Masson trichrome blue staining and alpha defensin immunostaining of low fibrotic score (**A**,**B**) and high fibrotic score patients at ×200 magnification (**C**,**D**) and at ×400 magnification (**E**,**F**). (**G**) fibrosis stage and the average alpha-defensin immunohistochemical staining at each score range. (*) indicates *p* value < 0.05.

**Table 1 medicina-59-00496-t001:** List of patients’ comorbidities at presentation expressed in %. HTN—hypertension, CAD—coronary artery disease, CHF—congestive heart failure.

	NAFLD Cases (*n* = 80)
Diabetes	37.84%
HTN	32.43%
Hyperlipidemia	47.95%
Obesity	50.70%
CAD	1.37%
CHF	1.37%

**Table 2 medicina-59-00496-t002:** List of patients’ baseline blood biochemical parameters at presentation expressed as median (IQR 1–3). HTN—hypertension, CAD—coronary artery disease, CHF—congestive heart failure.

	Fibrosis 0–2	Fibrosis 3–4	*p* Value
ALT (U/L)	65 (46–110.5)	56 (34–66.8)	0.03
AST (U/L)	42 (28–55.5)	39.5 (28.8–57.5)	0.56
LDH (U/L)	378 (337–457)	357 (324–379)	0.06
Albumin (g/dl)	4.6 (4.4–4.8)	4.5 (3.7–4.7)	0.01
CRP (mg/L)	3.5 (2.1–7.4)	6.8 (3.2–13.6)	0.047

## Data Availability

Not applicable.

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
