# Peer review of "Human Neutrophil α-Defensins 1–3 Are Upregulated in the Microenvironment of Fibrotic Liver"

_medicina, 2023, doi:10.3390/medicina59030496_

Round 1
Reviewer 1 Report
In this manuscript, Rami Abu Fanne and coworkers evaluate the expression of alpha defensins 1-3 in the liver of patients with low or high fibrosis in a NAFLD setting and is based on one main result.
Major comments:
1- While described in the results, the number of patients used for this study is not specified in the materials and methods
2- Please show the data as dot plots, rather than histograms to have a clear view of the data. In addition, median and IQR will also give a better idea of the data rather than mean and SD (including results of biochemical tests listed in text).
3- Results of metabolic and biochemical parameters (BMI, AST, ALT, Albumin, alkaline phosphates…) should be summarized in a table for an easier reading, and also probably distinguished according to the fibrosis groups (F0-F2 vs F3-F4)
4- What does the authors mean by biochemical hepatitis? On which biochemical parameters is based this definition of hepatitis? Please clarify. I don’t think that this term is appropriated since hepatitis is determined by histological examination.
5- In the results section, authors repeatedly mention “significant” differences between the two tested groups. However, figures 1 and 2 does not mention any significant results between F0-F2 and F3-F4 groups. Please correct the text and the figures. Please mention statistics in the figures. Only figure 3 mention its statistics. Statistics should be included in all figures.
6- In fact, the only main message of the paper is the accumulation of alpha defensins (presumably secreted by neutrophils) around fibrotic areas. Is-it due to increased neutrophils infiltration or activity? An additional experiment of neutrophils staining using specific markers by immunohistochemistry is mandatory, to define whether this increased alpha defensins in fibrotic livers is due to an increase of neutrophils infiltration or to an increase in neutrophils activity.
A colocalization analysis of neutrophils and alpha defensins stainings would be appreciated.
7- It is stated in the materials and methods section that: “Comparisons were then made between the three different stains to evaluate correlation/colocalization between alpha-defensin immunostaining and steatosis/fibrosis in each specimen”. Where are associations of alpha defensins with steatosis? Authors associated alpha defensins levels to fibrosis only. Please correct this issue.
8- Additionally, I didn’t see in this study any correlation analysis. Please correct “correlation” by “association”.
9- Image 1: Pictures of F0-F2 group (A and B) seems to be at a different magnification than pictures of F3-F4 group (C and D). Please show representative images at the same magnification between groups.
10- References n°5 and 25 are not appropriated, please correct that.
Minor comments:
1- References n°1 to 3 in the introduction section should be replaced by more recent studies (such as ref 23)
2- Reference n°12 in the material and methods section should be replaced by a more appropriated reference, i.e. the original description of NAS score (Kleiner, Hepatology, 2005; PMID 15915461)
3- A list of abbreviation has to be made.
4- Author contribution as well as fundings of the study should be stated.
5- In the introduction section, it would be appreciated to include a sentence stating whether alpha defensins levels in the liver have been already associated with liver fibrosis (including in other contexts than NAFLD, in animal models…)
6- The histological quantification method of alpha defensin staining is not clear. Did authors quantify the whole slides? Or did they quantify a number of images randomly taken for each patient? If yes, the number of analyzed images, the magnification used for the analysis, should be said in the methods. Please clarify.
7- Abbreviations used if Table 1 have to be detailed in the legend of the table.
8- Image 1 and Figure 3 (quantification of image 1) should be pooled in the same figure
9- The color code of Figure 3 should be harmonized with the one of fig 1 and 2 (F0-F2 in grey; F3-F4 in black).
10- In Image 1, pictures magnification should be mentioned, at list in the image legend.
11- In the discussion section, it is written: ““Similarly, in the current study steatosis was inversely related to fibrosis level (Figure 2)…”. There is no significant difference in steatosis between the two groups, and not even a trend toward less steatosis in F3-F4 group. Please remove this sentence.
12- This sentence in the discussion is not clear, please correct: “In line with previous reports, insulin resistance correlates, the strongest risk factor for NAFLD and the predisposition of individuals to cirrhosis, were more prevalent in patients with more advanced fibrosis”
Author Response
Review Report Form
Answers to reviewer 1-
In this manuscript, Rami Abu Fanne and coworkers evaluate the expression of alpha defensins 1-3 in the liver of patients with low or high fibrosis in a NAFLD setting and is based on one main result.
Major comments:
1- While described in the results, the number of patients used for this study is not specified in the materials and methods
Answer- patients number was added in page 1.
2- Please show the data as dot plots, rather than histograms to have a clear view of the data. In addition, median and IQR will also give a better idea of the data rather than mean and SD (including results of biochemical tests listed in text).
Answer- Technically the statistician could not provide a suitable dot plot graph…
We provided the median (IQR) values instead of the mean (SD).
3- Results of metabolic and biochemical parameters (BMI, AST, ALT, Albumin, alkaline phosphates…) should be summarized in a table for an easier reading, and also probably distinguished according to the fibrosis groups (F0-F2 vs F3-F4)
Answer- the results of the biochemical parameters were attached in new table, table 2, page 7, and they were distinguished according to the fibrosis group.
4- What does the authors mean by biochemical hepatitis? On which biochemical parameters is based this definition of hepatitis? Please clarify. I don’t think that this term is appropriated since hepatitis is determined by histological examination.
Answer- "biochemical hepatitis" was meant to represent "hepatic injury"/abnormal liver enzymes indicated by elevated serum ALT levels. The term "biochemical hepatitis" was accordingly changed in page 7.
5- In the results section, authors repeatedly mention “significant” differences between the two tested groups. However, figures 1 and 2 does not mention any significant results between F0-F2 and F3-F4 groups. Please correct the text and the figures. Please mention statistics in the figures. Only figure 3 mention its statistics. Statistics should be included in all figures.
Answer- indication of significance was attached in all graphs, and also in the text whenever relevant.
6- In fact, the only main message of the paper is the accumulation of alpha defensins (presumably secreted by neutrophils) around fibrotic areas. Is-it due to increased neutrophils infiltration or activity? An additional experiment of neutrophils staining using specific markers by immunohistochemistry is mandatory, to define whether this increased alpha defensins in fibrotic livers is due to an increase of neutrophils infiltration or to an increase in neutrophils activity.
Answer- using the H&E, fibrosis, and alpha defensin staining, the pathologist confirmed a heterogeneous staining for released alpha-defensin in the absence of neutrophils in association with tissue scaring (see a representative image with magnification in image 1 (E&F).
7- It is stated in the materials and methods section that: “Comparisons were then made between the three different stains to evaluate correlation/colocalization between alpha-defensin immunostaining and steatosis/fibrosis in each specimen”. Where are associations of alpha defensins with steatosis? Authors associated alpha defensins levels to fibrosis only. Please correct this issue.
Answer- no association with steatosis was confirmed. A correction was made in page 6 and the word steatosis was deleted.
8- Additionally, I didn’t see in this study any correlation analysis. Please correct “correlation” by “association”.
Answer- thank you for this valuable remark. The word correlation was changed to association.
9- Image 1: Pictures of F0-F2 group (A and B) seems to be at a different magnification than pictures of F3-F4 group (C and D). Please show representative images at the same magnification between groups.
Answer- images with similar magnification were provided in image 1.
10- References n°5 and 25 are not appropriated, please correct that.
Answer- thanks again for the accurate revision. Referenced 5 and 25 were replaced with appropriate ones.
Minor comments:
1.References n°1 to 3 in the introduction section should be replaced by more recent studies (such as ref 23)
Answer- more recent studies were included instead of the previous 1 to 3 references.
2- Reference n°12 in the material and methods section should be replaced by a more appropriated reference, i.e. the original description of NAS score (Kleiner, Hepatology, 2005; PMID 15915461)
Answer- reference 12 was changed with a more appropriate one.
3- A list of abbreviation has to be made.
Answer- a list of abbreviation was attached in page 3.
4- Author contribution as well as fundings of the study should be stated.
Answer- author contribution and fundings were provided in pages 11 and 12.
5- In the introduction section, it would be appreciated to include a sentence stating whether alpha defensins levels in the liver have been already associated with liver fibrosis (including in other contexts than NAFLD, in animal models…)
Answer- at the bottom of page 4 there is a sentence stating a previous work confirming association between non-physiological levels of alpha defensin and liver fibrosis: " Similar findings were described earlier (11) when pan-tissue expression of alpha-defensin increased hepatic fibrosis by inducing hepatic stellate cell proliferation, with a null effect on hepatic steatosis".
6- The histological quantification method of alpha defensin staining is not clear. Did authors quantify the whole slides? Or did they quantify a number of images randomly taken for each patient? If yes, the number of analyzed images, the magnification used for the analysis, should be said in the methods. Please clarify.
Answer- it was clarified in the head of page 6:
"Quantitative analysis
Immunohistochemical reactions were carried out on three sections. Five randomly selected microscopic fields (each field 0.785 mm2, 200 × magnification) from each liver section were then photographed with a digital camera (Olympus C-4000, Japan). The data were subjected to software-assisted image analysis (ImageJ®) for quantification of immunohistochemical staining intensity".
7- Abbreviations used if Table 1 have to be detailed in the legend of the table.
Answer- the abbreviations in table 1 were detailed.
8- Image 1 and Figure 3 (quantification of image 1) should be pooled in the same figure
Answer- image 1 and figure 3 were pooled in "figure 3".
9- The color code of Figure 3 should be harmonized with the one of fig 1 and 2 (F0-F2 in grey; F3-F4 in black).
Answer- the color code of Figure 3 was harmonized with the one of fig 1 and 2.
10- In Image 1, pictures magnification should be mentioned, at list in the image legend.
Answer- pictures magnification was mentioned in the legend of figure 3.
11- In the discussion section, it is written: ““Similarly, in the current study steatosis was inversely related to fibrosis level (Figure 2)…”. There is no significant difference in steatosis between the two groups, and not even a trend toward less steatosis in F3-F4 group. Please remove this sentence.
Answer- this "in-accurate" sentence was modified accordingly.
12- This sentence in the discussion is not clear, please correct: “In line with previous reports, insulin resistance correlates, the strongest risk factor for NAFLD and the predisposition of individuals to cirrhosis, were more prevalent in patients with more advanced fibrosis”
Answer- the sentence was corrected in page 11: "the traditional risk factors for NAFLD and the predisposition of individuals to cirrhosis- hypertension, hyperlipidemia, diabetes and obesity, were more prevalent in patients with more advanced fibrosis".

Reviewer 2 Report
In a retrospective study, the authors found that a higher metabolic risk was associated with a higher grade of fibrosis in NASH/NAFLD patients. And immunostaining for alpha-defensin showed patchy intense staining concordant with high fibrosis, nicely co-localized with histological fibrosis. I have a few suggestions to the authors of this work.
1. The meaning of * should be indicated in figure 3.
2. Another suggestion is that the statistical tests used in the data are not specified. For example, it should be stated whether the Mann-Whitney U test or the student t test is used for continuous variables. It should also be noted that Pearson's chi-square test or Fisher's exact test was used for categorical variables.
3. The results section has been kept very short. In this section, important parameters can be presented to the reader in a more understandable way. I suggest that this section be reviewed by the authors.
4. How these tests are taken from the patients in the laboratory, which tubes they are sent to the laboratory, and which devices these tests are used with should be stated in the material and method section.
5. In addition, were patients using anti-inflammatory drugs or patients with diseases that increase inflammation included in the study?
6. Is it possible for the author to quantify the content of alpha-defensin, and then analyze the correlation between the content of alpha-defensin and the fibrosis score by using pearson correlation, and show it with scatter plot.
7. Can the authors show the differences in baseline measures between the fibrosis score groups in the table?
Author Response
Answer to reviewer 2
Open Review
Comments and Suggestions for Authors
In a retrospective study, the authors found that a higher metabolic risk was associated with a higher grade of fibrosis in NASH/NAFLD patients. And immunostaining for alpha-defensin showed patchy intense staining concordant with high fibrosis, nicely co-localized with histological fibrosis. I have a few suggestions to the authors of this work.
1.The meaning of * should be indicated in figure 3.
Answer- the meaning was provided in figure 3.
- Another suggestion is that the statistical tests used in the data are not specified. For example, it should be stated whether the Mann-Whitney U test or the student t test is used for continuous variables. It should also be noted that Pearson's chi-square test or Fisher's exact test was used for categorical variables.
Answer- the statistical tests used were specified in page 6.
- The results section has been kept very short. In this section, important parameters can be presented to the reader in a more understandable way. I suggest that this section be reviewed by the authors.
Answer- thank you for this valuable point. Relevant modifications were done in the results section.
- How these tests are taken from the patients in the laboratory, which tubes they are sent to the laboratory, and which devices these tests are used with should be stated in the material and method section.
Answer- clarification was made in page 5: " Relevant biochemical parameters drawn just before each liver biopsy were retrieved from the Hillel Yaffe medical center archive. All tests were previously analyzed using the Cobas® 6000 analyzer series (Roche Diagnostics)".
- In addition, were patients using anti-inflammatory drugs or patients with diseases that increase inflammation included in the study?
Answer- in the 1st paragraph in page 5 we clarified that neither inflammatory medication was regularly used, nor overt systemic inflammatory disease included.
- Is it possible for the author to quantify the content of alpha-defensin, and then analyze the correlation between the content of alpha-defensin and the fibrosis score by using pearson correlation and show it with scatter plot.
Answer- unfortunately, the accessible data does not include a pinpoint matching of alpha defensin score and fibrosis score. We have a table that includes all alpha defensin values for fibrosis group (0-2) and another table for alpha values obtained in the fibrosis group (3-4). Consequently, we have the average alpha intensity in both the low and high fibrosis scores.
- Can the authors show the differences in baseline measures between the fibrosis score groups in the table?
Answer- differences in baseline biochemical measures were included in the new table 2, page 7.

Round 2
Reviewer 1 Report
Thank you for replying all requests.
Author Response
Thank you
Reviewer 2 Report
After the authors revised the manuscript, the language of the manuscript improved significantly. However, I believe that the clinical information of the study subjects in this manuscript has major shortcomings, which can not better reduce confounding factors between the groups. In addition, the experimental data is too less to better demonstrate the relationship between alpha defensin and fibrosis. The scientific nature of logical reasoning needs to be strengthened. Therefore, I don't think the manuscript is ready for publication.
Author Response
Modifications were done in pages 2 and 6